# Interactions between the Public and Assistance Dog Handlers and Trainers

**DOI:** 10.3390/ani11123359

**Published:** 2021-11-24

**Authors:** Bronwyn McManus, Gretchen Good, Polly Yeung

**Affiliations:** 1School of Health Sciences, Massey University, Palmerston North 4442, New Zealand; bmcmanus1@inspire.net.nz; 2School of Social Work, Massey University, Palmerston North 4442, New Zealand; P.Yeung@massey.ac.nz

**Keywords:** assistance dog, service dog, disability assist dog, disability, social model of disability, discrimination, denied access, social interactions

## Abstract

**Simple Summary:**

The number of disability assistance dogs in Aotearoa New Zealand (NZ) is small but slowly growing; therefore, encountering an assistance dog in a public place remains a novel experience for most people. Little is known about the experiences of NZ handlers and trainers with the public. In this research, there were both benefits and challenges for participants when interacting with the public. Handlers benefited from increased social engagement but they experienced the challenges of denied access to businesses, cafés, restaurants, shops, and public transport; invasive personal questions; unwanted interactions; and interference with the dog. These challenges were most difficult to manage. These findings highlighted the complexity of such interactions and the need to inform the public about the dog/handler or dog/trainer teams’ legal right of access to public places and etiquette on how to interact with these teams.

**Abstract:**

This research aimed to explore the experiences of handlers and trainers of disability assistance dogs in terms of the types of interactions they had with members of the Aotearoa NZ (NZ) public and how these interactions were perceived, interpreted, and managed. A qualitative method, guided by an interpretive approach and social constructionism, was utilised to collect data via semi-structured interviews with six handlers and six trainers of assistance dogs. Data were analysed using thematic analysis with the social model of disability as the theoretical base. Findings indicated that participants regularly faced a complex range of unique interactions due to various factors such as the public’s lack of knowledge and understanding of the dog’s role and right of access to public places. While participants encountered brief friendly comments about the dog and its role, other encounters involved long conversations, invasive personal questions, interference with their dogs, and denied access to businesses, cafés, restaurants, and public transport. These findings underpin the need to provide more education to the public on the etiquette of engaging with handlers and their assistance dogs and more support for businesses to understand the legal rights of handlers. Through education and support to change societal attitudes and remove structural barriers, disabled people using assistance dogs may be able to independently participate in community life and be fully included without hindrance.

## 1. Introduction

One of the core functions of disability support is providing appropriate resources to help disabled people live full and independent lives. Assistance dogs are a form of assistance that is growing in popularity. Disabled people choose to use an assistance dog for various reasons, such as reducing the impact of their impairment on their daily lives, increasing their independence, and accessing the community. Assistance Dogs International (ADI) [1] provides standardised training and certification protocol for dogs that assist disabled people. ADI states that assistance dog is a generic term covering three types of assistance dog: guide (dogs that guide visually impaired people), hearing (dogs that alert to sounds for hearing-impaired people), and service dogs (dogs that assist people with disabilities other than hearing or vision impairment). Assistance dogs are trained to undertake three or more tasks for disabled people [2]. They may assist, for example, with mobility [2]; opening doors; pressing buttons; picking up items [3]; alerting to sounds [4] or with medical alerts, such as seizure [5] or hypo- or hyper-glycaemia events [6]; helping to keep a child safe by preventing the child from running off; or helping a child to walk safely as part of a parent/child/dog triad team [7]. Although an assistance dog may be of great practical assistance, accessing the community can be fraught with social challenges because, at times, the dog becomes the primary focus of public attention. This extra attention from the public can distract the dog from its task and causes delays, stress, or embarrassment for the disabled handler as well as distracting the dog from its work and interfering with its training. Conversely, a secondary outcome of working with an assistance dog is that the dog may also create new opportunities for social engagement, assist in developing friendships, create a sense of inclusion, and positively change public perception of disabled people [7,8,9,10].

Previous research has established that the benefits to handlers of assistance dogs include increased independence, assistance with time and energy conservation, and improved quality of physical and emotional health [2,7,11]. Other studies discuss how the presence of a dog promoted social engagement with the public, as the dog created a common talking point, thus encouraging friendly interaction and conversation between the handler and members of the public [3,12,13,14]. Even though studies have highlighted the benefits of assistance dogs, handlers have frequently reported on social media that issues arise when taking their dogs into a public space [15]. National newspapers have reported that handlers of assistance dogs face many challenges, such as being refused access to public transport [16], motels [17], and restaurants [18], as well as the problems associated with members of the public interfering with guide dogs (dogs that assist blind people) [19]. It also appears that the public is largely unaware that their well-intended, friendly approaches, especially without asking for permission first, distract the dog from its training and ultimately its trained purpose. Repeated interactions and prolonged conversations can cause delays and be tiring for the adult handler, the disabled child who is part of the parent/dog/child triad, the trainer, or the dog [2,9,20,21].

The terminology used to define dogs that are trained to assist disabled people varies within and between countries. The terms service dog and assistance dog are used most frequently and interchangeably in NZ and abroad. NZ legislation [22] adopted the term disability assist dog rather than assistance dog but this term is rarely used by people working with these dogs; instead, assistance or service dog is used. To avoid confusion and maintain consistency with the international terminology, assistance dog is used in this paper and refers to guide, hearing, and service dogs.

Legislation varies among countries regarding the training and public access rights of assistance dogs accompanying their handler or trainer. In NZ, the Dog Control Act 1996 [22] (the Act) defines disability assist dogs (assistance dogs) as a dog that is in training or is trained by specified organisations to assist a disabled person. The organisations are named in the Act and subsequent amendments to the Act and orders [23,24]. The organisations must comply with the NZ guidelines for authorisation to certify disability assist dogs (assistance dogs) [25]. Under these guidelines for authorisation [25], the organisations must provide documentation to meet the following criteria: 1. Demonstrate that there is a need for the organisation’s dogs; 2. Demonstrate that the organisation has appropriate training practices and resources and support for the clients (handlers); and 3. Demonstrate that the organisation has an appropriate legal, management, and governance structure [25]. The organisation must apply to the Department of Internal Affairs with the appropriate documentation and if approved the organisation’s name is added to the relevant legislation. The organisations are required to be a member of and meet minimum best practice standards set by ADI or International Guide Dog Federation. The dog must wear an identifying jacket or harness and/or identification card, and meet the requirements for working in public places. At the time of this study, the following six organisations were authorised and certified: Hearing Dogs for Deaf People; Mobility Assistance Dog Trust; Blind Low Vision Guide Dogs; Assistance Dogs NZ Trust; Perfect Partners Assistance Dogs Trust; NZ Epilepsy Assist Dogs Trust. K9 Medical Detection NZ was granted certification after this study was completed and was not included. Only dogs trained and provided by these named organisations have the right of public access when working with their handler or trainer. If the dog is not working and does not have its jacket or harness on, it no longer has the right of access to public places and must abide by the same regulations governing dogs.

Businesses are expected to be aware of the public access rights of handlers and trainers accompanied by their working assistance dog. Unlike other countries such as the USA [26], NZ does not have legislation regarding what the public can ask a handler or trainer, but under the Dog Control Act 1996 [22], the handler or trainer must comply with any reasonable requests from the owner or occupier of the business or premises. By law, handlers and trainers of assistance dogs in NZ have the same obligations as all dog owners; to register, control, care for, and ensure the dog does not cause a problem for people, other animals, or property. Assistance dog training organisations in NZ generally state that handlers should keep their dog well groomed, clean, and well behaved in alignment with ADI best practices. In general, a handler or trainer and their working assistance dog cannot be denied access to any place the general public can access, such as buses, trains, taxis, rental cars, aeroplanes, cafés, restaurants, food halls, shops, shopping malls, supermarkets, town centres, cinemas, concerts, theatres, tourist places, hotels, motels, hospitals, public parks, and gardens.

The number of officially trained and certified assistance dogs in NZ is extremely small, with approximately 400 handlers in a population of just over 5 million people, with the largest number of dogs concentrated in the Auckland area. Therefore, these dogs are not only a new type of assistance for disabled people, but they are a relatively uncommon sight in public places, especially outside the Auckland region. Members of the public are generally unaware that refusing access to an assistance dog handler or trainer is illegal or of the impact denying access has on the handler. Such limited knowledge impacts upon the handler’s ability to go about their daily activities, resulting in a complex range of enabling and disabling interactions.

The purpose of this paper is to highlight the disabling/challenging issues that arise for trainers and handlers of assistance dogs when they interact with the public and the impact of such interactions.

## 2. Materials and Methods

### 2.1. Participants

Participants were recruited from the six certified organisations in NZ that trained and provided assistance dogs at the time of the study. Six trainers and six handlers volunteered to participate. The six handlers were made up of three parent handlers and three adult handlers. Written informed consent was obtained from participants before the interview. As the number of handlers and trainers of these dogs is very small in NZ (approximately 400 out of a population of five million), only general information about the participants is provided and pseudonyms are used to protect their identities. Participants included five female and one male handler, four female and two male trainers. Ages ranged between 20 and 65 years of age for both handlers and trainers. Participants had between 1 and 20 years of experience working with assistance dogs. Dog breeds included Labradors (3), Labradoodles (2), and Alsatian (1).

The six trainers represented five of the six Assistance Dogs International (ADI) certified assistance dog organisations in NZ. At the time of the study, one organisation did not have a trainer, and two trainers were from the same organisation. All trainers were experienced assistance dog trainers. Of the six trainers interviewed, one trainer’s primary role was training the dogs, and another’s leading role was instructing the handlers. The other four trainers had mixed roles, training dogs and instructing handlers. One trainer was also a disabled assistance dog handler. In this study, the term “trainer” refers to a person qualified to train the dog to the required standard. They also teach the new handler how to work with the dog and provide ongoing support throughout the dog’s working life.

The six handlers represented five of the assistance dog organisations. Three adult handlers stated that they had either vision impairment, physical impairment, or a medical condition. Two adult handlers were experienced and were working with their second dog (Labrador and Labradoodle). The other adult handler was a young adult with her first dog (Alsatian). Two handlers were first-time parent handlers, of whom each had a primary-school-aged child with a learning/behavioural impairment. In this situation, the child had impairment, but until the child reached the age of 16 years and was capable of independently working with the dog, a parent remained the handler and was responsible for controlling the dog and its care. Both families had Labradors. The third parent was initially a parent handler, but now her child was an adult who was able to work with the dog (Labradoodle) independently and did not wish to participate in the research. This parent also had many years of experience with respite care of dogs and puppy walking (early in-home training of puppies). She shared her experiences of outings with her child and the dog and when out on her own with an assistance dog. All the children lived with their parents. Five handlers lived in suburban environments, and one in a rural area. The term “handler” refers to the person who is in charge of the dog when it is working. This may be a disabled adult handler who works independently with the dog or a parent (parent handler) of a disabled child who is under 16 years of age or is unable to independently work with the dog.

### 2.2. Procedures

Ethics approval was received on 1st June 2017 from Massey University’s Human Ethics Committee: Southern A. Semi-structured interviews were conducted between June and September 2017 by B.M. The six training organisations were contacted and asked to forward, via email, an invitation to participate to their handlers and trainers. Handlers and trainers were asked to contact the researcher if they wished to participate. Thirteen people responded: six trainers and seven handlers. One handler replied late and was not interviewed. A total of 12 people were interviewed. Handlers included three parent handlers and three adult handlers. Ten participants (five trainers and five handlers—three parent and two adult handlers) were interviewed by telephone and two (one adult handler and one trainer) face to face. Interviews lasted between 45 and 90 minutes. Participants were asked to speak openly about any interactions they had with the public when out with their dog in a public place. All interviews were transcribed verbatim by BM Data were analysed using the six stages of thematic analysis outlined by Braun and Clarke [27,28]. Data were grouped under five themes: denied access to shops, cafés, and restaurants, denied access and other issues on public transport, unwanted interactions between the public and the dog/handler or trainer team, the human–animal bond, and the roles of ambassador and educator.

The social model of disability and social constructionism was the theory base. Social constructionism adheres to the belief that knowledge and reality are socially constructed based upon society’s cultural, religious, political, and social norms [28,29]. This perspective is used to seek understanding of how people perceive, seek meaning, interpret, and understand their physical, emotional, and spiritual world [30]. The social model fits within this paradigm as it is based upon the concept that disability is socially constructed [31,32,33]. The social model highlights how society creates disabling barriers based on rules, attitudes, and beliefs that exclude disabled people from participating in everyday activities across environments. It also highlights how society can remove these barriers, leading to inclusive and enabling environments [31]. New Zealand legislation and government strategies for disabled people are based on this social model and have begun to address the physical barriers but research indicates that disabling social attitudes and beliefs have yet to change [26,34,35].

## 3. Results

Participants stated that the majority of interactions with the public were positive, with most people treating them respectfully. Some handlers enjoyed the increased attention the dog attracted, as without the dog, the public ignored or avoided them. When convenient, participants were willing to discuss with the public the dog’s role and purpose and allow people to pat their dog. Even though most interactions were positive and enabling, it was the unwanted and disabling interactions the participants discussed in great detail as these interactions caused the most distress for both the participant and their dog. As the beneficial and enabling interactions are covered in the current literature, it is the disabling or challenging interactions that are the focus of this paper. Pseudonyms were used for participants.

### 3.1. Denied Access to Shops, Cafés, and Restaurants

Disabling interactions occurred when shop, café, or restaurant staff acted as gatekeepers and refused the handler or trainer access due to the presence of their dog. Participants reported receiving comments such as, “No you’re not allowed in here with a dog, get out” (Tasha, trainer) or “you’re not allowed pet dogs in here” (Hayley, adult handler). When participants explained they did have legal access and produced printed evidence of the Dog Control Act, staff provided other reasons to decline entry such as the following: “dog smells” or “our clients could be allergic to dogs” (Trevor, Trainer). One trainer was threatened with “I will call the police” (Tasha, Trainer).

Trainers had greater confidence in negotiating access as it was part of their professional role to gain unobstructed access for the next handler or trainer and provide education to staff. Whereas handlers, especially novice handlers, found addressing denied access difficult as they had less confidence in managing such confrontations. Heather, a parent handler, stated that her teenage child, when learning to work with his dog independently, was refused access to the local dairy. Heather said: “[My son] went up there with the dog and was told he couldn’t have the dog in there and [he] came back home without whatever he had been asked to go up and get.” More experienced handlers had the confidence to ignore the request to leave “[they asked] me to remove the dog please and I’ve said no it’s a working dog and I’ve just ignored them” (Hayley, adult handler).

Handlers reported receiving assistance from others when access was denied, either from family who were with them or other members of the public. Heather followed up the dairy incident by going to the shop herself and explaining that her son and his dog did have right of access. In other situations, café patrons have intervened and said “he is so allowed in here if you don’t let him stay we are going” (Heather, parent handler).

When access is reluctantly granted, sometimes participants are made to feel unwelcome and receive poor service. Hayley, an adult handler, stated that: “I will not go back to that restaurant because the atmosphere was very cold”. Trevor, a trainer, gained entry but was told to sit outside or by an open window because the dog smelled. As it was winter they chose to leave that café and find a place that did accept them. On the other hand, perseverance paid off. When Tasha, a trainer, was refused entry to a café, she ignored the request to leave and the threat to call the police and instead she sat down and waited. She stated:

“I placed our order [and sat down] … after about 20 minutes, she [waitress] came over and said ’I didn’t realise that the dogs were that good I was expecting your dog to be wandering around’, and I said thank you for the compliment.”

To avoid conflict and a distressing situation when access was denied, handlers indicated that they were advised by the assistance dog organisation to obtain a name and contact details and politely leave. The handler was then asked to pass the information to the organisation so that either a trainer or CEO of the dog’s training agency could contact the business to provide education on the handler’s legal public access rights. Although this process was suggested to handlers, no handlers said they had followed it when access was denied. Instead, handlers reported that they managed denied access themselves even though these situations were distressing. Teagan, a trainer, said they were aware of how distressing disabling encounters were for handlers, so they thought this approach helped reduce the distress and embarrassment of such encounters for disabled handlers. She stated:

“The secondary stuff that goes on for them [handlers] such as anxiety that would cause them in situations to become really, really stressed out, which puts their health at risk…They just end up feeling like they should just hide or leave, and that’s not nice.”

Passing the issue onto the organisation to manage may reduce the handler’s distress and embarrassment, but it does not necessarily make the next denied access situation any more manageable and does not empower the handler to address the issue. Hence, this action may decrease independence to access places handlers want to go and maintain dependence on non-disabled people to negotiate access on their behalf, thus counteracting the dog’s initial purpose.

### 3.2. Denied Access and Other Issues on Public Transport

Two handlers reported difficulty gaining access to buses and one parent handler reported her son had problems on the train. Hanna, an adult handler, had repeated issues gaining access to the bus. Hanna reported, “A bus driver said, ‘It’s only Guide Dogs [that are allowed on]’, and I can’t have my pet on the bus. I said, “I’ve got the law right here if you want to look at it.’” Hanna stood her ground, and with support from companions, she eventually gained access after the driver called the depot to check if the dog was allowed on the bus. However, in the meantime, Hanna felt embarrassed by delaying others from getting on the bus and having her disability exposed. After her third denied access incident on a bus, she stated “if I had another issue, I’m not going to be polite, I’m going to be rude.”

Helen, a parent handler, gained access to the bus but then had an upsetting incident with a passenger. She said once she was on board the bus:

“I get a little tap on my shoulder, and I turned round and there’s this guy in his 30s. He goes, ‘dogs aren’t allowed on the bus’. I said, ’Oh, this one is; it’s a working dog.’ He proceeded to have this big argument with me about dogs not being allowed on the bus. Other people were asking him to let it go and leave [us] alone. He just persisted the whole time.”

This incident was so distressing for Helen that she had not attempted to use a bus again. The incident resulted in her having to explain to her daughter about being disabled in front of a public audience. She was also concerned about how her daughter would manage such incidents in the future when she was able to be an independent handler.

Heather described similar incidents when her adult child started to independently use the train with his assistance dog. She said, “initially, he used to have trouble on the trains with people telling him he couldn’t have the dog on the train; it was always the conductor.” This resulted in her contacting the transport manager and providing education on her son’s right of access when accompanied by his dog.

In both situations, the transportation staff member said that they did not believe the handler even when evidence was provided, and it took the support and intervention of non-disabled people for the disabled handlers to be granted access that they rightfully had. To resolve the issues of access to public transport, both Hanna and Heather took further action of making a complaint to the management to obtain some resolution. Heather said, “it got resolved after a few letters were sent to the people—management… [in a] senior role there, and he sorted it out well and truly.” Hanna’s mother also complained, resulting in getting “free bus rides out of that, which we weren’t expecting; we got a formal apology saying our policy is you can come on.” For Hanna, it was apparent that the staff were not informed about public access rights of handlers and trainers of assistance dogs, as over the following months, she continued to be refused access. The lack of ongoing education from management to existing and new staff continues to create barriers for disabled handlers, resulting in unnecessary stress, frustration, and the need to fight constantly for the right of access.

### 3.3. Unwanted Interactions between the Public and the Dog

Participants described situations where they were ignored and the person intentionally attracted the dog’s attention or rushed up to the dog and began patting. Helen, a parent handler, summed up these experiences by saying:

“People calling out ’doggie doggie’, trying to using the high pitched voice, whistling, and people deliberately block [ing] our path to get the dog’s attention...Then putting their head down trying to make eye contact with the dog and then we have got people running up to us ’Oh, I know I shouldn’t touch the dog but I can’t resist’ and then they touch…The final thing is often people engage with the dog almost as if we are invisible.”

In other situations, people were more stealthy and would try to surreptitiously sneak a pat of the dog as they passed by without asking the participant’s permission. Hanna, an adult handler, said:

“My Mum and I would call it ‘drive-by petting’. It’s when they [public] are just parallel, and they would stick their hand out. I’m not touching, but it’s just there, and it’s just grazing past. It’s like; you are still petting the dog.”

Trainers also experienced unwanted interactions during training sessions which disrupted the dog concentration on the task being taught and prolonged the session. Dog breeds such as Labradors and Golden Retrievers, most often used as assistance dogs, are naturally highly sociable and teaching a dog to ignore strong social cues was difficult. Tanya, a trainer said:

“It does depend on the type of dog and how much people push that engagement. A little, quiet pat on the head is unlikely to disturb them but then you get some people that do the high pitched voice ‘Oh he’s gorgeous’ and make eye contact and really trying to engage the dog. When the dog engages back that’s when what they’re s’pose to be doing sort of goes out the window.”

Repeated unwanted interactions with the dog during training or handover to handler had the potential to teach the dog to stop every time a person approaches rather than remain focused on the commands from the handler. Tina, a trainer, noted; ”stop, start, stop, start, stop, start becomes the normal for the dog”. She further reiterated that:

“When you stop the dog for no reason what you’re doing is saying look at the members of the public because they might talk to you and be nice to you rather than we’ve stopped because of a work situation or we’ve stopped because there is a curb or an obstacle. What you’re doing is encouraging the dog to look for attention.”

Peoples’ lack of awareness of the dog’s presence resulted in potential injury to the dog. This issue appeared to be a problem in crowded places such as supermarkets. Helen, a parent handler, said: “just to give you an example our dog’s been hit in the head with a basket full of wine bottles, she’s had her tail run over, and her feet trodden on.” Helen stated such action from the public was distressful, saying: “Just because people are in a big rush and I don’t know if they can’t see us or what but they just don’t care and they don’t say sorry or anything. It’s awful.” Another adult handler, Hanna, noted how well her dog tolerated being trodden on. “Someone accidentally stepped on his foot. He didn’t move, he didn’t make a noise and they are like ‘Oh my God he didn’t do anything he didn’t yelp’. I said, ‘yeah that’s how good they have to be.’”

Participants noted that unwanted or unexpected interactions could be stressful for the dog. When Helen, a parent handler, noticed drive-by petting of her dog, she stated, “she [dog] was really holding it together and I thought you are just a superstar. That’s really stressful for the dog.”

In another situation, Tasha, trainer and handler, and her dog were unexpectedly surrounded by a group of excitable young school children wanting to touch her dog. She reported:

“You could see that he was feeling stressed and trapped…but there was no way out. I just said to them ’don’t talk to him back, back, back my dog needs some space’. One of the parents said to me ’but my child wants to pat your dog’ and I said, ‘excuse me I said I’m sorry but the needs of my dog has to come first, your child can pat any old dog’.”

The dogs are trained to ignore such strong stimuli and not react to unwanted attention. As Tanya, a trainer, stated:

“If you just go up and pat a dog without any sort of introduction of the dog, the dogs do amazingly well to put up with that but the other part of that is if the dogs didn’t, all hell would break loose.”

### 3.4. Human–Animal Bond

Handlers reported they developed a close bond with their dog such that the dog was an extension of themselves. When people touched their dog, handlers said it was the same as a stranger touching them, which was intrusive and an unwanted invasion of their personal space. Helen, parent handler, explained, “I often feel like when somebody touches [dog] it’s like somebody touching my child. Why are you touching, you haven’t asked me. It doesn’t belong to you. It’s quite an invasion.”

An adult handler, Holly, stated, “when you’re interacting with the public, they don’t perceive you as a private citizen.” She concluded that the public did not regard disabled people as private individuals; therefore, members of the public felt entitled to report, comment, criticise, and touch them and their dog without respect or regard for the handler’s privacy or the dog’s welfare. Such lack of respect for the strong bond between the handler and their dog resulted in the handler being objectified. There is a lack of awareness that the dog and its handler are a single unit with a strong bond and that disabled handlers are capable of competently working with and caring for their dog.

### 3.5. The Roles of Brand Ambassador and Educator

Being an educator and brand ambassador was part of the trainers’ role for which they were trained. For handlers, these were additional, unexpected, and complex roles. Participants reported they were aware that when they were out in the community, they represented not only their dog’s training agency but also the other assistance dog organisations. This study’s results indicate that the general public appeared to lack knowledge that guide dogs were not the only type of assistance dog. Teagan, a trainer, stated:

“We are in a public education and the public limelight a lot…and we have to be mindful of that at all times, not just for ourselves but for everyone—Guide Dogs, Assistance Dogs, Hearing Dogs…No matter what brand we have on us we all, unfortunately, get put in the same category…People just don’t understand the differences. A lot of us smaller organisations, new organisations, we still get mistaken as Guide Dogs.”

The lack of understanding about the different types of assistance dogs resulted in handlers and trainers feeling that they represented all of the assistance dog organisations. Holly, a disabled handler, referred to this as being a “brand ambassador.” The brand ambassador role places additional responsibilities onto the handlers. The trainers and organisations provided some support, and some handlers cultivated their own ways of managing these additional roles.

The brand ambassador role placed the additional responsibility of always being polite and respectful towards the public even in the most difficult situations. Thus, by taking an assistance dog into a public place, the participants felt the need to be well prepared, tidily dressed, and polite to ensure they portrayed the overall brand of assistance dog positively and made a good impression. At the same time, the dog also needed to be clean, tidily groomed, and very well behaved. Being polite and respectful was, at times, difficult, but something participants felt obliged to maintain for the sake of the brand and impression their behaviour would have on other handlers. Being polite and respectful was also necessary; as Holly, an adult handler, stated, “people will report you if you aren’t—they’re quite willing to call the council and all sorts.”

Participants were aware that all the organisations were charities and dependent on donations from the public to assist with funding. Tanya, a trainer, acknowledged this dependence and stated: “we are, I think, more reliant on the generosity of individuals, small companies or individuals and fundraising and friends now. I think that we’re a lot more reliant on that to keep these charitable services going.” Reliance on public funding was another reason why maintaining the assistance dog brand in a positive light and always being polite seemed important to participants.

Educating the public was a role that handlers undertook casually during an interaction and more formally when giving talks to different community groups. Most handlers accepted this role even though it was an extra, unexpected responsibility over and above the purpose of having their dog. For handlers, this role required additional personal effort to prepare for the interactions. Hanna, an adult handler and student, provided her lecturers with written instructions on interacting with her and her dog to be shared with other students and staff, and she gave a talk to her class. Heather, a parent handler, introduced her son and his dog to the ambulance staff. Heather said, “I took him [dog] up to the local ambulance station and talked to them…They know that the dog has to go in the ambulance with them.” Participants reported that introduction and education sessions were necessary to ensure handlers had easy access to frequently used places.

Providing education to the public, formally or informally, required the handlers to be well prepared, have a good knowledge of assistance dogs, and be confident enough to talk to the public. It was a skill that developed with age and experience. These roles required confidence, time, and energy for handlers, of which disabled people and families with disabled children often did not have a lot to spare. In Holly’s words, handlers needed to “be prepared, practised and polished” in their engagement with the public.

These results indicate a total imbalance in the dissemination of information, with the responsibility for the education of the public and businesses remaining firmly in the hands of the handlers, trainers, and the assistance dog organisations. Constantly engaging with individual members of the public required a lot of time and effort on the part of disabled people, who, at times, have limited energy for such interactions. Businesses do not appear to be complying with their legal obligations or making an effort to inform their staff.

## 4. Discussion

The results of this study indicated that public attitudes towards the presence of trainers and disabled handlers and their assistance dogs in the public environment created complex and contested situations that are both enabling and disabling for disabled handlers and trainers. Based on reports of their interactions with the public, the following key issues were identified. The participants were all knowledgeable about the dog’s role and purpose, they knew their legal right of access, were willing to be a brand ambassador and educator, and remained polite and respectful towards the public even in the most difficult situations. On the other hand, the public seems to have limited knowledge and understanding of: the complexity of disability; the role and purpose of assistance dogs; the close relationship between the dog and its handler; the etiquette of engaging with the dog and its handler or trainer; and the legal rights of the dog when working with its handler or trainer in public places, especially, in shops, food outlets, and public transport. The public’s general lack of knowledge impacted participants’ ability to go about their daily activities, creating complex situations that were both enabling and disabling, depending on the context of the encounter. For handlers, no matter how well prepared or experienced they were, each encounter was unique and posed issues, with some being easier to manage than others. These encounters were influenced by, for example, time constraints, number and length of encounters, invasiveness of questions, and level of politeness or rudeness of the public.

Sometimes, limited public knowledge resulted in the dog being a social lubricant or ice breaker, drawing the public to the disabled handler as it created a common point of conversation. For parent handlers and some adult handlers with visible disabilities, such interactions were welcomed as they provided an opportunity for social interaction which did not occur when the dog was not present. Such interactions created feelings of inclusion for the handlers and opportunities to provide education about both the dog’s role and disability. Such findings are frequently reported in the existing literature [3,8,36,37,38]. In contrast, trainers and some adult handlers, especially those with invisible disabilities, found such interactions unwanted and an invasion of their privacy, supporting Mill’s findings [26]. Complexity arose as each interaction was uniquely dependent on factors such as time restraints, dog and handler’s energy levels, number and length of encounters, invasiveness of questions, and level of politeness or rudeness of the public. One of the most frequently encountered disabling issues all handlers and trainers faced was being denied access to businesses, such as shops, cafés, restaurants, and public transport. Participants stated that staff said the dog was not allowed access for reasons such as the dog was dirty, smelly, might steal food or frighten customers, or staff just did not want the dog on the premises. Such discriminatory attitudes and behaviours demonstrated a lack of understanding by business staff that the dog with its handler or trainer had a legal right to access. These experiences have been documented in other studies concerning denied access to shops [10], cafés or restaurants [39], and accommodation [40,41,42].

Even though the right of public access policy and the law varies among countries, studies in Sweden, Estonia, Germany, the US, and Japan [39,40,41,42] indicate that a lack of knowledge of the law or policy on the part of business personnel is a universal problem. This study has further highlighted that businesses in the community still struggle to facilitate inclusive access to disabled people and their assistance dogs. Participants reported that business managers had not ensured that their frontline staff knew the legitimate right of access; they were not aware of how to identify an assistance dog and did not realise that denying them access was illegal, discriminatory, and distressing. Existing research [26,40,41] emphasises that it is imperative that business staff and personnel fully understand the legitimate right of access of the handler or trainer when working with their assistance dog. It is essential that staff are conversant with legitimate reasons for excluding the handler or trainer and their dog. When lack of knowledge was combined with a misuse of the gatekeeping role—namely, denial of access—this created a power imbalance that placed disabled handlers in a position of continually needing to be prepared to defend their legal right of access or walk away from the situation (as recommended by the training organisations). To walk away from a situation meant the handler submitted to people’s discriminatory actions, resulting in them giving up their right to: access the community; use public transport or taxis; or attend social functions, a restaurant, a shop, an event, or an attraction of their choice. If the handler departed, leaving an unresolved situation due to the business personnel’s inappropriate actions, then nothing had changed. The business personnel may continue to be ignorant of the fact that their actions were discriminatory, disabling, or potentially in breach of the handler’s human rights [43].

What has become apparent from this study is that although the NZ law protecting the rights of disabled people regarding access to public places has been in place for over 20 years, handlers in this study have reported ongoing issues and continue to be discriminated against and ostracised by disabling societal attitudes. Similar issues have been reported by participants in existing studies [9,26,44,45]. The sobering reality depicted in the current research further illustrates that while the rights of disabled people have changed in areas such as education, health, and employment, the rights of those who use assistance dogs have shown minimal practical change. The limited level of knowledge, understanding, and acceptance of the access laws by business personnel has shown minimal progress and disabled people are not yet valued as equal citizens. Until business personnel take responsibility for educating their frontline staff, these significant barriers to inclusion will remain, and NZ assistance dog handlers will continue to be forced to educate businesses or continue to endure discrimination.

This study also identified that handlers were the first point of contact for the public to gain information about the dog’s role and disability. Even though at times handlers found interactions with the public discriminatory and disrespectful they submitted to such behaviour, as being a good brand ambassador for the dog training organisation and disabled people was prioritised over their own needs. All participants wanted more public education related to appropriate etiquette for interacting with disabled people and their assistance dogs. This could enable them to participate in daily activities without unwanted interference and the perceived need to always be polite and well presented. Mills [26] suggested a need for a public education campaign to inform people that an assistance dog was no different from any other assistance for disabled people and that they should ask before touching the dog, a view supported by this and other studies [9,20,36,37,46,47].

This study identified that the bond between the handler or trainer and their dog is strong. Handlers often view their assistance dog as an extension of themselves. This study indicated that the public may be unaware of this close bond that forms a single unit. So rather than seeing a single unit, the public sees a person and a dog, therefore, treating them separately. When strangers are compelled to interact inappropriately with a working assistance dog, it demonstrates a level of disrespect for the handler and invades the handler or trainer’s personal space. It also creates a dilemma for the dog as even the best trained dog can find such enticing stimuli very difficult to ignore. These types of actions distract the dog from its purpose of assisting its handler and can disrupt and prolong its training. All these situations can cause distress for the dog and its handler or trainer and at the same time highlight the unique role of the assistance dog in assisting disabled people.

When the public is unaware of the dog’s presence, the dog may be at risk of injury. Again, this creates a dilemma as the dog, on one hand, is not meant to solicit attention and should blend into the background but on the other, people need to be aware of its presence to avoid injuring it. What is apparent from this study is that participants are cognisant of the dog’s welfare but at times limited public awareness of the dog’s presence places its safety at risk. Furthermore, the lack of acknowledgement of injuring the dog distresses the handler, creating yet another dilemma regarding choosing to address the incident or ignoring it.

This research identified that the NZ public lacks knowledge and understanding of assistance dogs’ roles, their right of access to public places, and etiquette of engagement with the dog and its handler or trainer. In addition, there is a lack of awareness of the complexity of disability in that impairment can be visible and invisible. At present, the role of educating the public is placed firmly in the hands of the handler or trainer and at times this role is unwanted, inappropriate, or inconvenient. Therefore, it is essential that business, transport, and other managers realise that the right of access for assistance dogs is not a company policy but a law by which they must abide. Business managers need to take responsibility for ensuring that their frontline staff are fully conversant with the laws surrounding assistance dogs and human rights in order to break down the structural barriers faced by disabled people. Such actions would enable staff to provide exemplary service and be proud of their knowledge and the service they provide [47]. Furthermore, the public need to be informed about how to engage with the dog/trainer or dog/handler team. Etiquette is simple—ask first and do not be offended if the answer is no. This recommendation was suggested previously by Spence [9]. Other actions include distributing posters or leaflets of the different types of assistance dogs with a brief statement of the laws to businesses and their frontline staff, including dog registration or city council documents, or placing them in shop windows. Such simple actions would assist with including disabled handlers in the community.

### Future Research

This study has identified that there is a lack of public knowledge about disability and assistance dogs which creates disabling barriers. Other research has identified other issues such as fear of dogs and cultural and religious norms that discourage interactions with dogs and disabled people. There is a need to examine these issues within the NZ context. Cross-cultural research may identify if the same attitudes exist across a range of countries and cultural, religious, ethnic, and social groups [48]. Further research is warranted to understand the acceptance of assistance dogs within the disabled population as this group is also ethnically, culturally, and socially diverse.

## 5. Conclusions

This study explored the experiences of 12 NZ assistance dog handlers and trainers and their interactions with the public. Many beneficial interactions were reported which enabled disabled handlers to feel included in society and helped break down barriers they faced. These positive interactions are frequently reported in other studies, supporting the advantages of having an assistance dog. This research has identified that lack of public knowledge about the dogs; the complexity of disability; laws governing the dog’s legal right of access; and etiquette on engaging with the participants persists and continues to create disabling barriers for disabled people. This limited level of knowledge demonstrates minimal progress towards accepting disabled people as equal citizens. Until these barriers to inclusion are addressed, handlers will continue to endure discrimination and be responsible for educating the public on appropriate and respectful ways of engaging with disabled handlers and their assistance dogs.

## Data Availability

The datasets for this article are not publically or otherwise available due to issues of confidentiality.

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
