# Peer review of "Interactions between the Public and Assistance Dog Handlers and Trainers"

_animals, 2021, doi:10.3390/ani11123359_

Round 1

Reviewer 1 Report

I commend authors for tackling a field of study very often ignored and understudied in our field, which is the disadvantages of having an assistance dog. This research is significant as it addresses the disadvantages and discriminations experienced by assistance dog handlers in a community. 

I have a couple of small recommendations for minor revision, especially in the methods which is currently missing many critical details. 

General: 

Although the term “disability assist dog” is appropriate for NZ legal implications, I recommend using the accepted terminology “assistance dog” in this manuscript as this is an international journal with international readership. Doing so will greatly increase the visibility and relevance of this paper to the assistance dog/service dog field. In addition, the “disability assist dog” adds unnecessary terminology to a field already distraught with confusion. Even though this is the term used in in NZ, the authors can state that the term assistance dog was used to align with international terminology ((using mobility assistance dog if you want to designate that they are for mobility purposes rather than guide/hearing etc)

Introduction: Authors cite the 2012 Winkle et al systematic review, but this should be replaced and/or supplemented by the most recent 2020 systematic review in this area (Rodriguez, K. E., Greer, J., Yatcilla, J. K., Beck, A. M., & O’Haire, M. E. (2020). The effects of assistance dogs on psychosocial health and wellbeing: A systematic literature review. PloS one, 15(12), e0243302).

Methods: “The social model of disability social constructionism was the theory base.” – This needs elaboration, description of theory and rationale, and citations.

Please state the gender/age/ethnicity and race of participants, either in the text or in a demographics table. 

There is also no information given on the dogs themselves - what were their breeds, ages, etc?

Please state recruitment numbers in terms of how many participants were contacted and how many agreed to be in the study. 

Results - It would be helpful to have longer quotes justified in the text to help readability, in it’s current format it is hard to know what is a quote and what is author text. 

Author Response

Response to Reviewer 1:

Thank you for the very useful feedback.

We acknowledge your suggestion that the introduction and methods sections require improvements. We have endeavoured to address each of your suggestions.

  • We have changed our terminology, as suggested, and are now using the more commonly used term, “assistance dog”.
  • We have now included, in the discussion, the updated reference you suggested: (Rodriguez, K. E., Greer, J., Yatcilla, J. K., Beck, A. M., & O’Haire, M. E. (2020. The effects of assistance dogs on psychosocial health and wellbeing: A systematic literature review. PloS one, 15(12), e0243302).
  • We have provided some elaboration in our description of theory and rationale, and we have provided references to support this.
  • We have now provided some basic demographic details about participants, but we must provide minimal information, because in our small country, participants could be recognized if too much detail is provided.
  • Further details about recruitment numbers have now been added.
  • The reviewer has asked us to make quotes easier to identify.  We have used standard APA (7) formatting; block quotations for 40 or more words and quotation marks around shorter quotes.  We have now added quotation marks to block quotes, as we have found this style within the Animals journal.  We hope you find this acceptable. 

Reviewer 2 Report

I am concerned about the terminology in this paper- disability assist dog. There is agreement in most regions that there are types of ASSISTANCE dogs: guide dogs, hearing dogs and service dogs (jobs other than visual and hearing). A new term is being introduced here which will make the likelihood of this paper being found in a search much smaller, and not readily usable as evidence for any of the above listed categories as it does not appear to be a frequently used term in the world of assistive technology nor in use with people who have disabilities that have been placed with either a guide, hearing or service dog. If all categories are being referred to, then the term ASSISTANCE DOGS may be more appropriate. I do understand that NZ has adopted nontraditional terminology, consider moving it closer to the beginning of the paper and then perhaps include parenthetical reference "assist (assistance) dog" or "assist (guide, hearing or service).

The first paragraph of the intro could use some references for the kinds of work dogs do (example: Assistance Dogs International).

By definition, most laws, and training standards, guide, hearing and service dogs are trained to perform tasks for people with disabilities. Psychosocial outcomes are not traditionally defined as that is not an assistance dogs job (please know that I agree that it is a GREAT outcome and quite important in the big scheme of things). Most laws outline that the dogs are to perform trained tasks- that is part of what separates them from a pet. They are not trained for psychosocial outcomes. There is a decent amount of argument about this amongst assistance dog trainers, law makers and researchers.

Many would say that looking at these areas may not be a worthwhile endeavor since this is not their job. Others would argue that it is an unplanned secondary outcome. The laws have been built to separate pets from assistance dogs so that people without disabilities do not abuse the laws or try to pass off their dogs as assistance dogs. For example, look at the chaos in the US with the blurred line of Emotional Support Dog ("more than a pet but not a trained service dog") for a period of time, they were allowed on aircraft and in public until it got so out of control that it became a public safety issue as these untrained dogs were harming others- dogs and people).  So please consider the approach of this paper and consider the exact words of your laws defining assistance (guide, hearing or service) dogs. Consider this for the title as well- make sure your paper comes up in searches for Assistance dogs (guide. hearing and service).

Assistance Dogs International is not exclusive to the US. Please correct so it include all other regions or simply INTERNATIONAL. See https://assistancedogsinternational.org/members/regional-chapters/

From the point of the discussion to the end was very well developed. I did wonder if some of the points from the discussion would be better in the introduction. In the end, I did understand the paper and several of my questions were answered. But I do think the terminology, definitions and descriptors should be in the beginning rather than the discussion. It appears that the NZ laws use different definitions and do not include responsibilities of handlers/trainers??? Consider putting that in intro early on.  Handlers keeping dogs clean and groomed/free of odor is usually a requirement of the handler/trainer for public access- so if your laws do not cover that and organizations are not pushing that- it is a worthwhile discussion (referring to the person who was asked to sit by a window as the dog smelled).

Otherwise, great topic and content!

Author Response

Response to reviewer 2:

Thank you for the helpful suggested revisions. We note that you suggested the introduction, results and conclusions sections could use improvements and we have reviewed these and made the revisions suggested.

  • As suggested by both reviewers, we have changed our terms from “disability assist dog” to “assistance dog”.
  • The first paragraph of the introduction contains multiple references and descriptions about the type of work assistance dogs perform. We hope you find this acceptable.   
  • We have carefully explained the work of assistance dogs in New Zealand and noted that the psychosocial outcomes are akin to unplanned secondary outcomes, utilizing references from NZ law and ADI resources.
  • Also, as recommended, we have moved the terminology, definitions, and descriptors to earlier sections of the manuscript rather than in the discussion.
  • We have acknowledged that ADI is an international body.
  • And a brief discussion about responsibilities of handlers/trainers has now been included in the introduction.
